# Kinetic, Isotherm, and Equilibrium Investigation of Cr(VI) Ion Adsorption on Amine-Functionalized Porous Silica Beads

**DOI:** 10.3390/polym14102104

**Published:** 2022-05-21

**Authors:** Anzu Nishino, Ayane Taki, Hiromichi Asamoto, Hiroaki Minamisawa, Kazunori Yamada

**Affiliations:** 1Major of Applied Molecular Chemistry, Graduate School of Industrial Technology, Nihon University, 1-2-1 Izumi-cho, Narashino, Chiba 275-8575, Japan; cian20012@g.nihon-u.ac.jp (A.N.); ciay22022@g.nihon-u.ac.jp (A.T.); 2Department of Basic Science, College of Industrial Technology, Nihon University, 2-11-1 Shin-ei, Narashino, Chiba 275-8576, Japan; asamoto.hiromichi@nihon-u.ac.jp (H.A.); minamisawa.hiroaki@nihon-u.ac.jp (H.M.); 3Department of Applied Molecular Chemistry, College of Industrial Technology, Nihon University, 1-2-1 Izumi-cho, Narashino, Chiba 275-8575, Japan

**Keywords:** porous silica beads, aminosilane coupling agent, polyethylenimine, XPS, hexavalent chromium, adsorption, water purification

## Abstract

The hexavalent chromium (Cr(VI)) ion adsorption properties were conferred to porous silica beads by introducing alkylamine chains through functionalization with an aminosilane coupling agent, [3-(2-aminoethylamino)propyl]triethoxysilane (AEAPTES), or with an epoxysilane coupling agent, (3-glycidyloxypropyl)triethoxysilane (GOPTES), and polyfunctional amine compounds or poly-ethylenimines (PEIs). The presence of amino groups on the silica beads was confirmed by XPS and the amount of amino groups increased to 0.270 mmol/g by increasing the AEAPTES concentration and/or reaction time. The adsorption capacity of the silica beads functionalized with AEAPTES was the maximum at the initial pH value of 3.0 and the initial adsorption rate increased with an increase in the temperature. The adsorption capacity increased with an increase in the amount of amino groups at pH 3.0 and 30 °C. The adsorption behavior obeyed the pseudo-second order kinetic model and was well expressed by the Langmuir isotherm. These results support that Cr(VI) ion adsorption is accomplished through the electrostatic interaction between protonated amino groups and HCrO_4_^−^ ions. In addition, the adsorption capacity further increased to 0.192–0.320 mmol/g by treating the GOPTES-treated silica beads with triethylenetetramine, pentaethylenehexamine, or PEI. These empirical, equilibria, and kinetic aspects obtained in this study support that the porous silica-based adsorbents prepared in this study can be applied to the removal of Cr(VI) ions.

## 1. Introduction

Chromium (Cr) is the seventh most abundant element on earth and naturally occurs in the environment in the Cr(III) and Cr(VI) oxidation states. The main anthropogenic sources include metallurgical industries, refractory brick production, electroplating, combustion of fuels, waste incineration, and the production of Cr-containing chemicals, mainly chromates and dichromates, pigments, Cr trioxide, and Cr salts. Natural sources include volcanic eruptions and the erosion of soils and rocks [1]. Cr(III) is generally regarded as less toxic and argued to be potentially therapeutic. On the other hand, Cr(VI) is a powerful oxidant and many of its compounds are very soluble in water. Cr (VI) is toxic and its effects are carcinogenic, mutagenic, and teratogenic in humans and animals [2]. Other sensitive noncancer effects of Cr(VI) compounds are, severe respiratory (nasal and lung irritation), gastrointestinal (irritation, ulcer of the stomach and small intestine), and hematological effects (microcytic, and hypochromic anemia), in addition, it can cause damage to reproductive organs and malfunctions, such as a decrease in sperm counts in males [3].

Conventional methods, such as chemical precipitation, ion exchange, reverse osmosis, coagulation, and adsorption, have been reported for the removal of Cr(VI) ions from aqueous media. Among these methods, adsorption is viewed as superior to other methods because of its simplicity, ability for regeneration, cost-effectiveness, and enabling large-scale applications. Moreover, the greatest advantage of the adsorption method is that by-products, such as sludge, are not generated [4].

Many adsorbents prepared from different origins including synthetic polymers, activated carbons, biomass, graphene oxide, nanoparticles, and biosorbents, have been investigated for Cr(VI) ion removal [5,6]. Porous materials have been also used as a matrix to generate new materials with improved functional groups (adsorption sites). Here, adsorption is usually limited by dominant functional groups present on the surface and within the pores of an adsorbent [7]. It is not a forgone conclusion that composite materials produce superior performance. Therefore, many researchers have strived to find suitable adsorbents for Cr(VI) ions with excellent adsorption capacity, selectivity, and/or fast binding kinetics while using the minimum dosage. This gives an indication that there is room for new research in terms of adsorbent constructions.

The construction of new adsorbent materials is a current requirement of great necessity. In recent years, mesoporous silica or glass beads have been used as an adsorbent matrix because of their high surface areas, adjustable size of pores, evident surface properties, good mechanical stability, and reduced toxicity [8,9,10,11,12,13]. Reactively active Si-OH bonds present on the surfaces of the pore walls can be modified with functional moieties. Porous silica beads are novel mesoporous materials with many intriguing properties, such as low bulk density, continuous porosities, high specific surface area, and extremely low thermal conductivity. Besides, they have the advantages of being robust against organic solvents, thermal stability and they are not capable of swelling. Because of these unique features, porous silica beads are a promising material as an adsorbent.

Here, we considered that an adsorbent for Cr(VI) ions with outstanding adsorption properties, such as a high capacity, a high efficiency, and a high rate, can be constructed from porous silica beads by utilizing their properties. Therefore, in this study, amino functional groups were introduced into the porous silica beads through functionalization with different aminosilane coupling agents, mainly [3-(2-aminoethyl-amino)propyl]trimethoxysilane (AEAPTES). The Cr(VI) ion adsorption behavior of AEAPTES-modified silica beads (AEAPTES-silica beads) was estimated as a function of the pH value, the temperature, and the amount of amino groups, and then kinetically and thermodynamically analyzed. In addition, a further increase in the adsorption capacity was attempted by the pretreatment of the silica beads with an epoxysilane coupling followed by functionalization with polyfunctional amine compounds or polyethylene-imines (PEIs).

## 2. Materials and Methods

### 2.1. Chemicals

The porous silica beads, Davisil Grade 636, (pore size: 60 Å, 35–60 mesh particle size, specific surface area: 480 m^2^/g) were purchased from Sigma Aldrich (Tokyo, Japan). As aminosilane coupling agents, AEAPTES (Tokyo Chemical Industry, Tokyo, Japan), N-(6-aminohexyl)aminomethyl-triethoxysilane (AHAMTES, Gelest, Morrisville, NS, USA), and 3-[2-(2-aminoethylamino)ethylamino]propyl-trimethoxysilane (AEAEAPTMS, Acros Organics, Geel Belgium) were used. An epoxysilane coupling agent, GOPTES, was purchased from Sigma Aldrich. Polyfunctional amine compounds, ethylenediamine (EDA), diethylenetriamine (DETA), triethylenetetramine (TETA), and pentaethylenehexamine (PEHA), were purchased from FUJIFILM Wako Pure Chemical (Tokyo, Japan). The chemical structure of the silane coupling agents used in this study was shown in Figure 1. PEI samples with a molar mass of 600, 1800, and 10,000 (0.6KPEI, 1.8KPEI, and 10KPEI) were also purchased from FUJIFILM Wako Pure Chemical(Tokyo, Japan).

### 2.2. Functionalization of Silica Beads

Before the coupling reactions, about 20 g of silica beads were washed by stirring in 6 M HCl for 1 h. After the silica beads were activated by stirring in 2 M HCl at 60 °C for 4 h, they were washed thoroughly with distilled water and then dried under reduced pressure [14,15]. Toluene was dried with a molecular sieve 4A for 2 days before use. Then, AEAPTES solutions were prepared at 0.25–0.5 M with dried toluene. Alkylamine chains were introduced to the silica beads by stirring 5 g of activated silica beads in 25 cm^3^ AEAPTES solutions for the prescribed times at 30 °C. The silica beads were washed with toluene, ethanol, and acetone several times, and then dried under reduced pressure [16].

In a similar manner, silica beads were functionalized with AHAMTES, AEAEAPTMS, and GOPTES at 1.0 M and 30 °C for 1 h. The polyfunctional amine compound solutions in 1,4-dioxane at 1.0 M and the aqueous PEI solutions at 10 wt% were prepared. The GOPTES-treated silica beads (GOPTES-silica beads) were mildly stirred in the polyfunctional amine compound solutions for 6 h or PEI solutions for 24 and 72 h at 70 °C to introduce amino groups on their surfaces [17,18].

### 2.3. Determination of Amount of Amino Groups

The silica beads functionalized with the aminosilane coupling agents (NH-silica beads) and GOPTES-silica beads functionalized with the polyfunctional amine compounds or PEIs were immersed in a 20 mM NaOH solution so as to neutralize the protonated amino groups on the alkylamine chains and then washed with water several times. Then, 0.2–1.0 g of the NH-silica beads were immersed in 20 mL of a 20.0 mM HCl solution and the solutions were mildly stirred for at least 24 h under a nitrogen atmosphere. The HCl solutions were titrated with a 20.0 mM NaOH solution with bromothymol blue as an indicator [3,19]. The amount of effective amino groups, A_NH_, was calculated from the concentration difference in the HCl solution using Equation (1):(1)$$A_{NH}^{}\;(\mathrm{mmol}/g)=\frac{(20.0-C)\cdot 0.020}{W_{NHSB}}$$
where W_NHSB_ is the weight of the NH-silica beads immersed.

In addition, the amount of alkylamine chains introduced to the silica beads was determined by dividing the A_NH_ value by the molar mass of an alkylamine chain (129.25 for AEAPTES, 185.36 for AHAMTES, and 172.32 for AEAEPTMS).
(2)$$A_{alkylNH}^{}\;(\mathrm{mmol}/g)=\frac{A_{NH}}{M_{\mathrm{alkylamine}}}$$

### 2.4. XPS Measurements

The XPS core level spectra of C_1s_, O_1s_, N_1s_, and Si_2p_ were recorded under reduced pressure lower than 5 × 10^−5^ Pa on a Shimadzu ESCA 3400 X-ray photoelectron spectrometer (Kyoto, Japan) with the MgKα (1253.6 eV) source operating at 8 kV and 20 mA for the untreated and functionalized silica beads. The electron energy calibration was performed by taking the Au_4f_ core level peak at 83.8 eV as the reference.

### 2.5. Cr(VI) Ion Adsorption

The calibration curve of Cr(VI) ions was prepared at the isosbestic point of 338.2 nm (r > 0.999, log ε = 3.454 dm^3^/mol·cm) [20,21]. The Cr(VI) ion adsorption was estimated by the following procedure unless otherwise described. The pH value of an aqueous K_2_Cr_2_O_7_ solution at 0.20 mM was adjusted to 3.0 with HCl. First, 10 mg of NH-silica beads were immersed in a pH 3.0 HCl solution for at least 12 h. Then, the NH-silica beads were placed in a 0.20 mM K_2_Cr_2_O_7_ solution (50 cm^3^) at 30 °C to initiate the adsorption experiments. The absorbance at 338.2 nm was measured at predetermined intervals to determine the Cr(VI) ion concentration. The adsorbed amount was calculated from Equation (3):(3)$$\mathrm{Adsorbed}\;\mathrm{amount}\;(\mathrm{mmol}/g)=\frac{(C_{0}-C_{t})\cdot 0.050}{W_{NHSB}}$$
where C_t_ and C_0_ are the Cr(VI) ion concentration at time t and the initial concentration, respectively, and W_NHSB_ is the weight of the NH-silica beads.

### 2.6. Cr(VI) Ion Desorption

Aqueous solutions of NaCl at 0.50 M, NH_4_Cl at 0.10 M, and NaOH at 0.50 mM were prepared as an eluent for desorption of Cr(VI) ions [4,20]. The Cr(VI) ion-loaded AEAPTES-silica beads were immersed in the above eluents at 30 °C. The absorbance at 338.2 nm was measured at predetermined intervals and the desorbed amount was calculated from the Cr(VI) ion concentration in the eluents.

## 3. Result and Discussions

### 3.1. Functionalization with AEAPTES

The silica beads were functionalized with AEAPTES by varying the concentration and reaction time. In this study, AEAPTES was selected as a main aminosilane coupling agent on the basis of the fact that a stable five-membered cyclic intermediate for intramolecular catalysis can form through the coordination of a secondary amine group to a Si atom for AEAPTES [22,23], indicating the hydrolytic stability of the AEAPTES-functionalized surfaces. This is an important feature because they are used in an aqueous medium. The amount of amino groups introduced was determined by the back titration with HCl. The amount of HCl consumed by protonating amino groups on the alkylamine chains was directly proportional to the amount of NH-silica beads added to the HCl solution (Appendix A). This indicates that the amount of introduced amino groups can be quantitatively determined by the back titration with HCl. Therefore, the amount of amino groups introduced to the silica beads was determined from the slope of the straight line. Figure 2 shows the changes in the amount of amino groups with the AEAPTES concentration at the reaction time of 1 h and with the reaction time at the AEAPTES concentration of 1.0 M. The amount of amino groups increased with an increase in the concentration and reaction time and/or the AEAPTES concentration and reached to 0.270 mmol/g at 1.5 M and 2 h.

In many studies on the functionalization of silica beads and glass beads with aminosilane coupling agents, the presence of introduced amino groups was confirmed by FTIR and XPS analysis. However, little was reported on the determination of the amount of amino groups [24,25]. Therefore, the determination of the amount of introduced amino groups is of great importance to discuss the Cr(VI) ion adsorption behavior stoichiometrically.

### 3.2. XPS Analysis

The strong peaks at 1080 cm^−1^ and at 805 cm^−1^ assigned to the stretching vibrations of the Si-O-Si bond were observed for AEAPTES-silica beads [20,21]. However, since no additional peaks by functionalization with AEAPTES were observed, the XPS analysis was also performed. Figure 3 shows the C_1s_, N_1s_, O_1s_, and Si_2p_ core level spectra for (a) untreated silica beads and (b) AEAPTES-silica beads. For the AEAPTES silica beads, in addition to a main peak, which was composed of a peak at 284.3 eV assigned to C-Si and a peak at 285.0 eV assigned to -CH_2_-CH_2_-, a small peak assigned to -C-NH- and -C-NH_2_ at 286.5 eV was observed in the C1s spectrum [26,27]. A N_1s_ peak assigned to -NHR- and -NH_2_ emerged and the O_1s_ and Si_2p_ peaks decreased. These results confirmed that amino groups were introduced on the surfaces of the silica beads according to the reaction scheme shown in Figure 1 [8,22,23].

### 3.3. Cr(VI) Ion Adsorption

The time course of the adsorption of Cr(VI) ions on the AEAPTES-silica beads with different amino group amounts at 30 °C and pH 3.0 is shown in Figure 4. There were fewer adsorbed Cr(VI) ions on untreated silica beads. On the other hand, for the AEAPTES-silica beads, the amount of adsorbed Cr(VI) ions increased against the immersion time and then reached equilibrium. This emphasizes that the functionalization with AEAPTES confers greater Cr(VI) ion adsorption performance to the silica beads by introducing amino groups on their surfaces. The initial adsorption rate was calculated from the slope of the linear region of the amount of adsorbed Cr(VI) ions against the immersion time and the half-adsorption time, t_1/2_, which is defined as the time required for the adsorption to take up half as much Cr(VI) ions as its equilibrium values, was also calculated [28].

#### 3.3.1. pH and Temperature Dependence

The effects of the pH value and temperature on the Cr(VI) ion adsorption were estimated for the AEAPTES-silica beads with the amount of amino groups of 0.220 mmol/g. Figure 5 shows the changes in the adsorption capacity, initial adsorption rate, and t_1/2_ value with the initial pH value. The adsorption capacity and initial adsorption rate had the maximum values at pH 3. The t_1/2_ value was at most 12.4 min at pH 3.0, indicating that the adsorption behavior is relatively fast.

Next, the adsorption behavior will be explained in terms of chromium species present in water [4,29,30,31]. In the range of pH 1.0–3.0, the equilibrium pH value was almost the same as the initial pH value within ±0.02. At pH 3.0, most of the Cr(VI) ions are present in the form of hydrochromate (HCrO_4_^−^) ions. When the pH value moves away from 3, the fraction of HCrO_4_^−^ ions decreases and the fraction of H_2_CrO_4_ increases. In addition, the concentration of Cl ions added to adjust the pH value increases. Meanwhile, the pH value at the adsorption equilibrium increased to 4.20 at pH 4.0 and to 5.28 at pH 5.0, respectively. In the pH range higher than 3, as the pH value increases, the fraction of CrO_4_^2−^ ions increases. In addition, an increase in the pH value results in an increase in the deprotonation of -NH_2_^+^- or NH_3_^+^ groups, because the dissociation constant (pK_a_) values of AEAPTES are 6.8 and 9.3 [32]. The above-described behavior is considered to result in a decrease in the adsorption capacity.

Based on the above results, the temperature dependence was investigated at pH 3.0. Figure 6 shows the changes in the adsorption capacity, initial adsorption rate, and t_1/2_ value with the temperature. When the temperature increased, the initial adsorption rate increased and the t_1/2_ value decreased. On the other hand, the adsorption capacity remained unchanged against the temperature. These results suggest that the Cr(VI) ion adsorption more successfully proceeded at higher temperatures. There are some systems in which the adsorption capacity is dependent on the temperature [31] and other ones where the adsorption capacity is independent of the temperature [3,4]. The effect of temperature on the adsorption behavior could vary depending on the experimental conditions, such as the type of adsorbent, amount and position of adsorption site, Cr(VI) ion concentration, adsorbent shape and so on. In addition, the activation energy will be calculated using the k_2_ values from the pseudo-second order equation in the following section to discuss the temperature dependence of the Cr(VI) ion adsorption in more detail.

#### 3.3.2. Amount of Amino Groups

At pH 3.0 and 30 °C, 10 mg of AEAPTES-silica beads with different amino groups were immersed in a 0.20 mM K_2_Cr_2_O_7_ solution and the adsorbed amount was measured against the immersion time. Figure 7a shows the changes in the adsorption capacity, initial adsorption rate, and t_1/2_ value with the amount of amino groups. The adsorption capacity and initial adsorption rate increased with the amount of amino groups and then leveled off higher than 0.12 mmol/g. In addition, the adsorption ratio was calculated by dividing the adsorption capacity by the amount of amino groups introduced on the AEAPTES-silica beads [3,4]. As shown in Figure 7b, the adsorption ratio gradually decreased with an increase in the amount of amino groups. It is found in Figure 4 that the adsorption reached equilibrium at relatively short immersion times; therefore, many of the alkylamine chains should be present on the surfaces of the silica beads. However, the initial adsorption and t_1/2_ value remained almost unchanged at higher amounts of amino groups.

In addition, to further increase the adsorption capacity, the functionalization of silica beads was also performed with other aminosilane coupling agents with two or three amino groups, AHAMTES and AEAEAPTMS. The adsorption capacity of AHAMTES-functionalized silica beads (AHAMTES-silica beads) and AEAEAPTMS-functionalized silica beads (AEAEAPTMS-silica beads) was summarized in Table 1. Although the amount of introduced amino groups for the AHAMTES-silica beads was the same as that for the AEAPTES silica beads, the adsorption capacity of the AHAMTES-silica beads was much lower than that of the AEAPTES silica beads. A low adsorption capacity for AHAMTES-silica beads would be due to the hydrophobicity of a hexylene chain between two amino groups in an alkylamino chain. On the other hand, the AEAEAPTMS-silica beads had a higher adsorption capacity than the AEAPTES-silica beads, since the alkyl chain length between amino groups is short (two ethylene chains) for the AEAEAPTMS-silica beads. The adsorption ratio based on an alkylamino chain was almost the same for the AEAPTES-silica beads and AEAEAPTMS-silica beads. These results support that the introduction of amino groups with AEAPTES was much more effective for the Cr(VI) ion adsorption.

#### 3.3.3. Kinetic Analysis

The adsorption data of the AEAPTES-silica beads with the amount of amino groups of 0.220 mmol/g were analyzed with the pseudo-first and pseudo-second order models. The pseudo-first order rate equation is generally expressed by:(4)$$\ln\;(Q_{eq}-Q_{t})=\;\ln\;Q_{eq}-k_{1}t$$
where Q_t_ and Q_eq_ denote the adsorbed amount at time t and at equilibrium, respectively, and k_1_ is the pseudo-first order rate constant.

On the other hand, the pseudo-second order equation is expressed as:(5)$$\frac{t}{Q_{t}}=\frac{t}{k_{2}\cdot Q_{eq}^{2}}+\frac{1}{Q_{eq}}t$$
where k_2_ is the pseudo-second order rate constant [9,33,34].

The kinetic parameters calculated by these two kinetic models were summarized in Table 2 and Table 3, respectively. The pseudo-first order equation fits only for the initial 12–20 min of the adsorption process. On the other hand, the pseudo-second order equation showed good linearity between the immersion time and the t/Qt value with higher regression coefficients for much longer times (Appendix A). These results indicate that the experimental adsorption data fit better into the pseudo-second order model than the pseudo-first order model. The good fit of the pseudo-second order model implies that this adsorption is of a chemical nature and the overall rate of the adsorption process is controlled by the rate-limiting step [35]. The k_2_ values were also calculated from the adsorption data at 20–50 °C discussed in the above section. Since the k_2_ values increased with an increase in temperature, the activation energy was calculated by plotting ln k_2_ against the temperature reciprocal.
(6)$${\ln\;k}_{2}=\ln\;A-\frac{E_{a}}{RT}$$
where A is the frequency factor, E_a_ is the activation energy, and R is the gas constant (8.314 J/K·mol). The activation energy was calculated to be 19.6 kJ/mol from the slope of the linear relationship of the ln k_2_ value with the reciprocal temperature (Appendix A). The activation energy for physical adsorption is usually not more than 4.2 kJ/mol. Therefore, this value suggests that the adsorption in this study occurs through a chemical reaction [36,37,38].

#### 3.3.4. Intraparticle Diffusion Model

Since the pseudo-second order model cannot identify the diffusion mechanism, the kinetic data were analyzed by the Weber and Morris intraparticle diffusion model to elucidate the rate-limiting step involved in Cr(VI) ion adsorption. The Weber and Morris intraparticle diffusion model [39,40] is expressed as:(7)$$Q_{t}=k_{id}\cdot \sqrt{t}+C$$
where k_id_ is the rate constant of intra-particle diffusion, t is the immersion time, and C is a constant that gives an idea of the thickness of the boundary layer.

Figure 8 shows a plot of the amount of adsorbed Cr(VI) ions against the square root of t for the AEAPTES-silica beads with the amounts of amino groups being 0.030, 0.080, and 0.220 mmol/g in a 0.20 mM K_2_Cr_2_O_7_ solution at pH 3.0 and 30 °C. The adsorption process was divided into three straight lines (k_id1_, k_id2_, and k_id3_) with regression coefficients higher than 0.99. The k_id_ values obtained were summarized in Table 4. The constant, C, corresponding to the y-intercept that provides the measure of the thickness of the boundary layer was not obtained, although it took 1 or 2 min for the adsorption behavior to reach the steady state. The first steeper region was observed for 20–30 min, indicating that this region is controlled by rapid external surface adsorption. The second linear region shows the gradual adsorption stage where the intraparticle diffusion is rate-limiting. The k_id_ values in this region were much lower than those in the first region (k_id1_ > k_id2_). The final third region is the equilibrium stage (k_id3_ = 0) [41].

#### 3.3.5. Isotherm Analysis

The adsorption capacity was determined in the range of the initial Cr(VI) ion concentrations of 0.05–0.30 mM at pH 3.0 and 30 °C for the AEAPTES-silica beads with the amount of amino groups of 0.220 mmol/g and applied to the Langmuir and Freundlich isotherm models, represented by Equations (8) and (9), respectively [35,41,42,43,44].
(8)$$\frac{C_{eq}}{Q_{eq}}=\frac{1}{K_{L}\cdot Q_{\max}}+\frac{C_{eq}}{Q_{\max}}$$
where C_eq_ is the Cr(VI) ion concentration at equilibrium, Q_max_ is the maximum adsorption capacity, and K_L_ is the Langmuir adsorption constant related to the affinity of binding sites.
(9)$${\log\;Q}_{eq}={\log\;K}_{F}+\frac{1}{n}{\log\;C}_{eq}$$
where K_F_ is the Freundlich constant related to adsorption capacity and 1/n is the empirical parameter corresponding to adsorption intensity, depending on the heterogeneity of the adsorbents.

The obtained parameters by both isotherms were summarized in Table 5 (Appendix A). The Langmuir isotherm model fits better with a higher regression coefficient than the Freundlich isotherm model. The good agreement with the Langmuir isotherm model was also confirmed by the fact that the calculated Q value is much close to the experimental one. These results suggest that the adsorption of Cr(VI) ion on the AEAPTES-silica beads depends on the concentration of the adsorbate. It can be discussed from the Cr(VI) ion species, that the adsorption process occurs mainly through the electrostatic interactions between a HCrO_4_^−^ ion and a protonated amino group as shown in Figure 13 in ref [4].

### 3.4. Desorption

The desorption of Cr(VI) ions was of great importance for the regeneration or reuse of adsorbents. Here, the desorption of Cr(VI) ions from AEAPTES-silica beads with the amount of amino groups of 0.220 mmol/g was measured in 0.50 M NaCl, 0.10 M NH_4_Cl, and 0.50 mM NaOH. These eluents were used for the desorption of Cr(VI) ions in our previous papers [4,19,45]. The desorption behavior was summarized in Table 6. Most or many Cr(VI) ions were released from the AEAPTES-silica beads. The desorption occurs mainly through the ion exchange of HCrO_4_^−^ ions with Cl^−^ ions on protonated dimethylamino groups, -NH^+^(CH_3_)_2_, in NaCl and NH_4_Cl and through the deprotonation from -NH^+^(CH_3_)_2_ to -N(CH_3_)_2_ in NaOH as shown in ref. [4] Figure 13, and [19,45]. Of them, the use of 0.10 M NH_4_Cl and 0.50 mM NaOH was effective for the desorption of Cr(VI) ions because high desorption % values were obtained, and the desorption was faster.

### 3.5. Functionalization with Polyfunctional Amine Compounds and PEIs

To further increase the Cr(VI) ion adsorption capacity, amino groups were introduced to the silica beads through the pretreatment with GOPTES and the subsequent functionalization of epoxy groups with polyfunctional amine compounds or PEIs. When the silica beads were treated with GOPTES, an overlapped small peak assigned to the -C-O- bond in an epoxy ring was observed at 287 eV as shown in Figure 3c. This indicates that glycidyloxypropyl chains were introduced on the surfaces of the silica beads. In addition, the GOPTES-silica beads were functionalized with four kinds of polyfunctional amine compounds and PEIs with three different molecular weights. As shown in Figure 3d,e, for the silica beads functionalized with EDA (EDA-silica beads) and with 0.6KPEI (0.6KPEI-silica beads) an N_1s_ peak appeared at 399 eV, indicating the introduction of alkylamino chains or PEI chains on the silica surfaces.

The C_1s_ peak at 287 eV and a N_1s_ peak was also observed for the functionalization with the other polyfunctional amine compounds and PEIs. Therefore, the amount of amino groups for these NH-silica beads was determined by the back titration, and then their adsorption capacity was measured at 30 °C and pH 3.0. As shown in Table 7 and Table 8, adsorption capacities higher than those of the AEAPTES-silica beads were obtained for the silica beads functionalized with the polyfunctional amine compounds and PEIs. In addition, the adsorption capacity increased to 0.201 mmol/g for the PEHA-functionalized silica beads (PEHA-silica beads) and to 0.320 mmol/g for the 0.6KPEI-silica beads. In particular, the adsorption capacity of the 0.6KPEI-silica beads was about 5.4 times higher than that of the AEAPTES-silica beads. These results support that the introduction of PEI chains or longer alkylamine chains to the silica beads is effective in producing a high-capacity adsorbent for Cr(VI) ions.

In addition, the Cr(VI) ion adsorption capacity of the NH-silica beads obtained in this study was compared with those of various particles functionalized by silane coupling reactions in Table 9 [8,11,12,13,46,47,48,49,50,51]. First, we should bear in mind that the experimental conditions, such as the initial pH value, temperature, dose of the adsorbent sample, and initial concentration and volume of the aqueous K_2_Cr_2_O_7_ solution, vary from article to article. However, it can be safely said that the adsorption capacity obtained in this study was comparable to, or higher than some of the other adsorbents, despite the Cr(VI) ion concentration being considerably lower than in the other articles cited. Our next goal is to prepare the adsorbent for Cr(VI) ions with better adsorption behavior through the functionalization with an epoxysilane coupling agent and polyfunctional amine compounds or PEIs.

## 4. Conclusions

In this study, the NH-silica beads were prepared through functionalization with different aminosilane coupling agents and through the pretreatment with an epoxysilane coupling agent; followed by the functionalization with polyfunctional amine compounds or PEIs. The kinetic, isotherm, and equilibrium characteristics of the Cr(VI) ion adsorption were investigated as a function of the experimental parameters, such as the pH value, the temperature, and the amount of amino groups in detail; desorption behavior was also discussed.

For the AEAPTES-silica beads, the adsorption capacity and initial adsorption rate had the maximum values at pH 3.0. On the other hand, the initial adsorption rate increased with an increase in the temperature, although the adsorption rate remained almost constant against the temperature. At 30 °C and pH 3.0, the adsorption capacity increased and the adsorption ratio decreased with an increase in the amount of amino groups. The kinetics of Cr(VI) ion adsorption obeyed the pseudo-second order model and the equilibrium data were well described by the Langmuir isotherm model. These results support that the Cr(VI) ion adsorption will occur through the electrostatic interaction between a HCrO_4_^−^ ion and a protonated amino group. Cr(VI) ions were successfully released in 0.50 M NaCl, 0.10 M NH_4_Cl, and 0.50 mM NaOH after a relatively short immersion time. In addition, the adsorption capacity was further increased by the pretreatment with GOPTES and the subsequent functionalization with PEHA to 0.201 mmol/g and with 0.6KPEI to 0.320 mmol/g, supporting that the introduction of PEI chains or longer alkylamine chains to the silica beads is effective in producing a high-capacity adsorbent for Cr(VI) ions.

The results obtained emphasize that the introduction of amino groups to the silica beads above-mentioned is an effective procedure to confer the Cr(VI) ion adsorption properties on the silica beads. The NH-silica beads exhibit significant potential as an adsorbent in the removal of Cr(VI) ions from aqueous media and wastewater. Therefore, we will aim for better adsorption behavior and higher adsorption capacity of Cr(VI) ions in the porous silica beads.

## Data Availability

All the data are available within the manuscript.

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
