# Peer review of "Kinetic, Isotherm, and Equilibrium Investigation of Cr(VI) Ion Adsorption on Amine-Functionalized Porous Silica Beads"

_polymers, 2022, doi:10.3390/polym14102104_

Round 1

Reviewer 1 Report

  1. The Novelty is not clear in the manuscript. The authors are suggested to make it more clear in the manuscript especially at the introduction section
  2. Technical writing is poor, and this area is more important to be refined in the manuscript before final approval
  3. Mechanism should be explained well to make the point clear in the manuscript
  4. Comparison with other materials in the form of table required

Author Response

We received e-mail and greatly thanked the reviewers in giving valuable comments for our manuscript. Here, we accepted many of the reviewer’s comments and revised the manuscript to publish in Journal, Polymers. Our responses to the reviewer comments are described as follows.

Answers to Reviewer 1

Comment #1 on novelty of this study

Reply: The explanation on use of porous silica beads to prepare an adsorbent for Cr(VI) ions is described in Line 64-73 in Introduction. In addition, some sentences were added in Introduction (Line 75-78) to make the aim of this study clear for readers.

Comment #2 on quality of English

Reply: So far, we have published more than 50 papers on surface modification and enhancement of adhesion and autohesion by photografting, removal of phenol compounds by the combined use of oxidoreductases and chitosan, development of polymeric adsorbents for removal of Cr(VI) ions in research journals with impact factors, such as Journal of Applied Polymer Science, Environmental Technology, Environmental Progress & Sustainable Energy, Journal of Polymers and Environment, and Polymers. Many related researchers have cited our Journal papers. Although our first language is not English, we can safely say that we have an adequate quality of English. However, we read the manuscript carefully and revised some parts.

Comment #3 on the mechanism of adsorption and desorption

Reply: We previously published some papers on adsorptive removal of Cr(VI) ions with polymeric materials prepared by photografting of an amino-containing monomer on PE and PTFE substrates. Therefore, additional explanations were described (Line 345 and 357).

Comment #4 on comparison with other adsorbents.

Reply: We compared the adsorption capacity obtained in this study with those of related particle adsorbents and presented Table 9 in the first draft. Our next purpose is to prepare Cr(VI) ion adsorbents through the functionalization by the combined use of GOPTES and PEI. Therefore, the comparison with other polymeric adsorbents is the next assignment.

Answers to Reviewer 2

Comment #1 on author names

Reply: The description of author names was revised according to the reviewer comment (Line 4).

Comment #2 on abstract

Replay: Some descriptions on quantitative results were added in abstract and abstract was revised (Line 3-30).

Comment #3 on superscript

Reply: This is a careless mistake. The number “2” was made superscript in Tables 2 and 3.

Comment #4 on unwanted letter

Reply: The unwanted letter “(6) was removed in eQ (7) (Line 310).

Comment #5 on the position of Figure 8 and Table 4

Reply: Since we revised the manuscript and added some new sentences in the content, the position of figures and tables was rearranged.

Comment #6 on the term “intraparticle”

Reply: We used the term “intermolecular” by mistake. Therefore, the term “intermolecular” was revised to the term “intramolecular” (Lines 305 and title of Table 4)

Comment #7 on regression coefficients of lines in Figure 4.

Reply: The typical results of intraparticle diffusion model are shown in Figure 8 and kid values are summarized in Table 4. Since there is not an adequate space in Table 4, a short description was added (Line 316-317).

Comment #8 on sentence of Line 343-345

Reply: We thank the reviewer for suitable comment. We revised the sentence according to reviewer comment (Line 338).

Comment #9 on supplementary materials

Reply: In supplementary materials, A correct and suitable graph was inserted in Figure S3 and the title was revised. The title of Figure S4 was also revised.

Reviewer 2 Report

The aim of the article is to present a research study regarding kinetic, isotherm, and equilibrium investigation of Cr(VI) ion adsorption on amine-functionalized porous silica beads. 

The Introduction section contains information regarding: (i) main anthropogenic source of Cr(VI) release to water; (ii) adverse health effects of Cr(VI); (iii) the choice of the adsorption process for the removal of chromium ion is justified; (iv) advantages of porous silica beads are presented.

The objectives are clearly stated.

Materials and Methods part contains six sections: (1) Chemicals; (2) The functionalization of silica beads; (3) Determination of amount of amino groups; (4) XPS analysis; (5) Cr(VI) adsorption. Cr(VI) ion desorption study is also presented.  All sections are described in sufficient detail in order to allow another researcher to reproduce the results.

In the Results and Discussion part, the authors present and interpret the results of the experiments performed. 

In the Conclusions section, the authors mentioned the specific results of their research investigation.

The paper is a well-written manuscript. Therefore, I suggest its publication after minor revision:

-Please remove ‘’and’’ between Hiromichi Asamoto 2 and Hiroaki Minamisawa 2 authors (Line 4)

-The Abstract section should contains some quantitative results/findings

-Table 2, Table 3, Table 5: Superscript ‘’r2’’

-Line 316: Please remove ‘’(6)’’ after ‘’C’’

-Figure 8 and Table 4 should be moved after Line 330

-Line 331: Please correct ‘’Interparticle’’ with ‘’Intra-particle”

-Table 4: Please correct ‘’Interparticle’’ with ‘’Intra-particle”

-Please add the value of correlation coefficient, R2, for Intra-particle diffusion model

-I recommend to improve the quality of the Figure 8

-Line 343-345: ‘’ The experimental data were analyzed in terms of the Langmuir and Freundlich  isotherms and the obtained parameters by both isotherms were summarized in Table 5  (Figure S4)’’ can be change with ‘’The obtained parameters by both isotherms were summarized in Table 5 (Figure S4)’’ (the information presented at Line 343 is already mentioned at Line 334)

-Please verify the Figure S3 (is identically with Figure S2)

-Please verify the legend of the Figure S4

-Line 391: Remove ‘’and’’ before “initial concentration’’

Author Response

(The authors gave the same response as above.)

Round 2

Reviewer 1 Report

I checked and the authors response to all questions and hence i recommend acceptance of the manuscript.